# Microscopic Structure from Motion (SfM) for Microscale 3D Surface Reconstruction

**DOI:** 10.3390/s20195599

**Published:** 2020-09-29

**Authors:** Dugan Um, Sangsoo Lee

**Affiliations:** 1Mechanical Engineering, Texas A&M-Corpus Christi, Corpus Christi, TX 78412, USA; 2Mechanical and Industrial Engineering, Texas A&M University-Kingsville, Kingsville, TX 78363, USA; sangoo.lee@tamuk.edu

**Keywords:** micro 3D imaging, 3D depth sensor, 3D shape reconstruction, photometric stereo, structure from motion

## Abstract

In microscale photogrammetry, the confocal microscopic imaging technique has been the dominant trend. Unlike the confocal imaging mostly for transparent objects, we propose a novel method to construct a 3D shape in microscale for various micro-sized solid objects in a broad spectrµm of applications. Recently, the structure from motion (SfM) demonstrated reliable 3D reconstruction capability for macroscale objects. In this paper, we discuss the results of a novel micro-surface reconstruction method using the Structure from Motion in microscale. The proposed micro SfM technique utilizes the photometric stereovision via microscopic photogrammetry. The main challenges lie in the scanning methodology, ambient light control, and light conditioning for microscale object photography. Experimental results of the microscale SfM, as well as the modeling accuracy analysis of a reconstructed micro-object, are shared in the paper.

## 1. Introduction

Three-dimensional depth-sensing technology has been a popular area of research in macroscale, as well as in microscale, due to scalable benefits in object recognition and identification. Three-dimensional depth imaging has enabled new levels of acquisition details in sensing not only by enhancing 2D imaging but, also, providing extra domain information for various applications. Compared to 2D vision, 3D vision is easier for shape analysis and more robust in object identification and classification due to the extra dimensional depth values. One of the most outstanding techniques in stereo-photometry is the surface reconstruction introduced by Woodham [1]. Woodham’s approach utilizes multiple images to build surface normal vectors. A 3D shape is, then, constructed through various Shape-from-X techniques. For example, shape-from-shading and the photometric stereo technique generate an elaborate 3D voxel of an object [2]. Another exemplary study includes the specularity of a high-reflectance material, such as a pearl [3]. The implementation into microscale applications such as a cell 3D structure, or micro-organ 3D imaging, however, are still dominated by confocal micro-imaging via blurriness due to the complexity in system configuration and problems in miniaturization [4]. 

The blurriness-based 3D micro-sensing technique is the most popular in various areas of micro-3D applications. In one article [5], the demonstrated idea was of identifying blurriness parameters around certain edges to generate a microscale 3D image. In another article [6], an original approach of a miniature scanning confocal microscope was described. The idea was to utilize MEMS (micro-electro-mechanical systems) technology for micro-3D vision, rather than using macroscale microscopy. The initial study in the paper coined the micro-vision via a micro-optical system, proposing further development toward the integration of several on-chip microscopes. Blurriness-based 3D imaging such as confocal microscopy or CLSM (confocal laser scanning microscopy) are important and have practical values. For instance, the reconstruction of a distortion-free ultra-high-resolution 3D model of a whole murine heart was demonstrated [7]. A distortion-free micro-3D surface was obtained by the multimodal registration of serial images generated by confocal laser scanning microscopy (CLSM) with the aid of a micro-CT (computerized tomography) 3D image as a template. Since the micro-CT did not provide information on the soft-tissue fine structure, a micro-CT image was used as a template to spatially co-register all the individual CLSM images. Cellular activity modeling via fluorescent imaging using laser scanning confocal microscopy was performed using dynamic imaging in conjunction with mathematical modeling [8]. A three-axis electromagnetic confocal micro-scanner was assembled for real-time imaging of vertical/horizontal cross-sections (relative to tissue) of biological samples [9]. Although a 50-µm-size onion cell structure was successfully constructed, the target vertical resolution was not achieved. This may be due to the limitation of the axial resolution of the laser scanning device. Reflectance confocal microscopy (RCM) has become popular in clinical applications. In order to shorten the image recovery process of RCM using compressive sensing, a deep-learning-based approach was studied [10]. It was informed that deep learning is useful, but application-specific training is required. Merging photoacoustic (PA) microscopy and fluorescence confocal microscopy (FCM) demonstrated extra-dimensional information for medical applications [11]. For instance, visualizing both blood and lymphatic vessels by PA-FCM enabled more comprehensive tracking of cancer metastasis. A Lissajous confocal endomicroscopic system was also developed for mouse tissue [12]. In the presented artifact, ex-vivo mouse tissue was obtained by the proposed system, opening opportunities for real-time fluorescence in-vivo applications in the future. In another artifact, a microwave was used to enhance confocal imaging for medical applications [13]. Although great progress has been made in confocal imaging technology, the primary disadvantage of the blurriness-based 3D imaging is the inability of reconstructing a micro-object with concave surfaces, especially a concave shape vertical to the scanning plane. In addition, a lengthy scanning time to process multiple layers at a time demerits the use of the confocal imaging system. For the most part, the high price tag on the imaging system for precision scanning control also leads to a big barrier for many users in various applications.

Structure from Motion [14] is a photogrammetric range imaging technique to generate a three-dimensional surface via the image stitching process. The Structure from Motion (SfM) technology reconstructs a 3D model by using motion parallax, which is the foundation of depth generation by measuring the amount of move of each feature as the camera moves. For instance, an object close to the camera moves faster than an object far away from it as the camera moves side-to-side. The fundamental mechanism of 3D construction is similar to that of stereovision, but photos next to each other form a pair of stereovision, thus enabling 3D reconstruction. Coined by the biological sensing of stereovision, it is now prevailing in various indoor and outdoor applications for 3D reconstruction. For instance, a technique to effectively manage impacts on property, infrastructure, and the broader ecosystem was presented by using Structure from Motion (SfM) photogrammetry [14]. Authors generated point cloud-based techniques for capturing the geometry and volµme of large wood accµmulations. Another area of SfM application is the rock slope discontinuity study. Although rock orientation has been traditionally measured through 3D laser scanning, the SfM technique, which is much more affordable, is becoming mainstream within the research community. However, it was noted that the SfM dataset showed inaccuracies on sub-horizontal and oblique surfaces [15]. In another case study [16], the 3D digital model of the Maddalena by Donatello was reconstructed via SfM. The paper described how the ICP (iterative closest point)-based alignment technique could lead to incorrect results in 3D surface accuracy. An alignment technique based on the fusion of ICP with close-range digital photogrammetry and a noninvasive procedure was also discussed to prevent the inaccuracy of the SfM process. Recently, several attempts were made in precision SfM for small artifacts such as a historical or cultural heritage [17,18]. In one article, 5–10-cm objects were 3D-modeled via SfM technology [17]. High-resolution imagery of a 3D-printed skull (5 cm) was achieved by using a 60-mm lens. Without a microscopic apparatus, however, the limit of the proposed SfM was for 5 cm or larger objects due to the camera focal distance limit and interference during the photography.

As we have seen in several examples, the SfM technique has become popular in various fields, mostly in macroscale. The SfM technology is known to be the most affordable technology for 3D reconstruction. No special hardware but a camera with reasonable resolution is required, since most of the 3D reconstruction is performed by the image stitching software. Although precision 3D modeling examples of small objects (5–10 cm) have been reported [17,18], a microscale SfM system has not been reported both in the industry as well as in academia due primarily to problems in miniaturization. Predominantly, the SfM technique, by its nature, requires photographing an object from different angles with adequate overlaps for the stitching process. Therefore, the main hurdle toward microscale SfM lies in the miniaturization of the SfM photographing method of a micro-object. In the next section, we discuss the important aspects of SfM technique and its principles from the perspective of how to downsize the macroscale methodology to a microscale 3D surface reconstruction process.

## 2. Toward Microscale SfM

Structure from Motion (SfM) is a 3D reconstruction technique by taking multiple 2D photography of an object from surrounding areas [19]. It is a photogrammetric range imaging technique for estimating three-dimensional structures from two-dimensional images by searching for common features or an object from different images. The assµmption behind the SfM is that a near object moves more than an object far away as the camera moves (see Figure 1). The original concept of the SfM is coined by the stereovision studied in the fields of computer vision and visual perception. In biological vision, SfM refers to the phenomenon by which hµmans can perceive 3D structure from projected 2D images at retinal areas. 

In general, SfM does not require a certain type of hardware or a special camera to generate a meta-shape of an object. However, a specific capturing sequence and guidelines are required for precision modeling. In this section, we discuss important factors that influence the accuracy of a 3D model by SfM technique and discuss challenging factors for downsizing the macroscale technology to microscale.

### 2.1. High-Resolution (5 MPix or More) Camera

First, Agisoft [20], one of the leading companies in SfM technique, recommends a high-resolution camera for an aerial or indoor 3D modeling process. The best choice for a common frame camera is 50-mm focal length (35-mm film equivalent) lenses. They also recommend a focal length from 20 to 80-mm intervals in 35-mm equivalents [21]. This requirement is difficult to accommodate in microscale SfM, since the focal length in microscale for an object below 500 µm is much less than that of a macroscale object. In addition, it is hard to generate a focused image for an entire micro-sized object in general. Therefore, in miniaturization, a microscope whose focal length is the largest in the class is required. In other words, unlike the confocal 3D imaging, micro-SfM would require a microscope whose focal length is larger than the depth of the measurement, thus preventing meta-shape distortion. In this paper, we used a microscope with 7×–45× zoom magnification (N.A.) power with 1-1/4” (33-mm) widefield of view and the working distance of 4″ (100 mm) [22].

### 2.2. Camera Calibration

If a dataset was captured with a special type of lens such as a fisheye lens, then the lens factor needed to be calibrated. This again is another barrier for a microscale SfM, since the photographing is not only taken by the camera lens but, also, through the lens assembly of the microscope. In general, most of the microscope uses a fixed lens; thus, we do expect that the calibration should not be as hard as is expected. The vendors of the SfM software usually require the focal length set either to maximal or to minimal value during the entire shooting session for more stable results. During the experiment, however, no significant difference was evidenced between the maximµm and minimµm values of the focal length. The result is due mainly to the fact that the focal length of the optical lens assembly of the microscope is different from the nominal value range of a camera.

### 2.3. Surface Texture

Another difficulty in microscale SfM is the inability of surface texture control. In macroscale SfM, care about the object’s texture is required. In addition, users need to invent tricks to avoid plain/monotonous and glittering surfaces. Generally speaking, a finely textured surface is more preferable than a shiny surface due to the meta-shape distortion by inconsistent light reflectance [21]. In the widely accepted Phong’s illµmination model, there are both diffusivity and specularity portions mixed in the light reflectance [23]. Phong’s illµmination model and photometry theory proposed that the light intensity, *I*, will be given by: The variables *C_o_* and *C*_1_ are two coefficients (diffusivity and specularity) that express the reflectivity of the surface being sensed, and *n* is a power that models the specular reflection for each material.
(1)$$I=C_{o}\left({\vec{\mu }}_{s}\cdot {\vec{\mu }}_{n}\right)+C_{1}{\left({\vec{\mu }}_{r}\cdot {\vec{\mu }}_{v}\right)}^{n}$$

Vectors *µ_s_*, *µ_n_*, *µ_r_*, and *µ_v_* are the light source, surface normal, and reflected and viewing vectors, respectively (see Figure 2). In Figure 2, θ stands for the angle between the reflected light and viewing angle. The vector *u_n_* is the normal to the object’s surface at the point of interest, *P* and *d* stand for the distance vectors (*l*) from each light source to the point, and *α* is the angle between *l* and the normal vector *N*. As expressed in Equation (1), diffusivity and specularity play an important role in the light reflectance measure, of which the accuracy of the photographic 3D modeling will be calculated. Inconsistency in the scene from different locations or angles will disturb the modeling accuracy, especially because of the feature-matching process of the image stitching. The portion that could produce more inconsistency in the light intensity measure due mainly to the different angle or location is the specularity term in the equation. While the angle *α* maintains constant during the photographing process, the angle θ will change, disturbing the intensity due to the change in the viewing angle. While diffusivity is less sensitive to the angle θ, the specularity changes significantly. There are materials that produce more specularity such as glass, metal, hermetically coated surfaces, etc.

In order to minimize the specularity in the light reflectance, surface preprocessing may be required. For instance, if the target object is a car, spreading some talc powder (or anything similar) over the surface is required to change from a glittering to a dull surface [21]. This mainly minimizes the effect of the specularity and maximizes the diffusivity to merit the field of view matching in multiple photographic images. 

In microscale SfM technology, however, the surface texture control is not feasible, thus making it challenging for downsizing the technology. In addition, ambient light control for microscale photography requires better control compared to the macroscale, further making microscale SfM difficult. We will discuss the ambient light control issue in the next section. 

### 2.4. Capturing Scenario

Another important factor in SfM technology is the capturing scenario to maximize the efficiency of the mathematical 3D photo stitching process. Since more than a 60% to 80% overlap between photos is required for the best modeling accuracy [21], a photography-capturing sequence must be designed in a way that the stitching process is able to extract enough matching features from neighboring photos. The best strategy is taking photos from 45 degrees around the vertical axis of the object, changing the viewing angle toward the center of the object. Multiple paths may be required to obtain a complete 3D model at different latitudes [21].

In addition to what we discussed so far, there are other factors in hardware configuration. For instance, the file format (TIFF preferable); EXIF data (sensor pixel size, focal length, etc.); ISO (lowest possible value); aperture (high enough for sufficient focal depth); and shutter speed (fast) are all a part of the factors. 

In sµmmary, below is the list of the most influential disabling factors identified for microscale SfM technology. 

Inability of surface texture change.Difficulties in ambient light control.Difficulties in capturing sequence generation.

In the next two sections, we will discuss the ideas of downsizing the macroscale SfM to microscale from two important perspectives of the SfM technology: capturing sequence and ambient light control. 

## 3. Photograph Capturing Sequence

As mentioned in the previous section, photographs of an object must be captured with a 60% to 80% overlap for the digital stitching process with 45 degrees around the rotational axis, pointing the viewing angle toward the center of the rotation. The sequence must be repeated with changing angles at a different latitude for the overlap in the upper and lower parts of the photos as well. 

However, the proposed sequence cannot be easily achieved in microscale, since the microscope and the camera assembly cannot move to generate a macroscale scanning pattern. In order to generate an optimal scanning pattern in microscale, we assembled a gantry-type microscope with a mobile platform underneath Figure 3.

The microscope supports an adjustable DOFo (Depth of Focus) and magnification capability of up to 45 times. The microscope and the photo-sequencing aperture were assembled on a vibration-isolated table for precision photography in microscale. A piezo-electric mobile platform with a rotational axis was assembled and placed underneath the gantry-type microscope (0.02 mm position accuracy). The digital camera attached on the microscope enables digital shuttling to cause no vibration during the photography. The proposed configuration provides solid rigidity during the stationary photography and enables testing various capturing sequences for microscale SfM. 

In order to obtain a complete set of scanning photography of a microscale object for the SfM process, we proposed a capturing scenario, as shown in Figure 4. The mobile platform underneath the microscope has two degrees of freedom axes of rotational mobility for varying latitude and longitude scanning. In addition, a three DOF (degrees of freedom) X-Y-Z motion control system enables precision focus of the micro-object for the microscope. The proposed capturing sequence is to ensure that the camera moves in order not to generate a blind spot or to better understand concave shapes on the surface, thus minimizing visual occlusion.

In addition, compared to the confocal micro-imaging technology (for instance, from Leica Microsystems), the scanning speed by the proposed micro-SfM technique is much faster. The scanning of an object with total of 30 photos by micro-SfM takes only 30 s, while the point scanning by confocal imaging with a x-y-z table for multiple images takes up to 30 min or more. The fast scanning speed of the micro-SfM is because of the simple scanning procedure with two rotational axes of the scanning platform.

## 4. Ambient Light Control

Unlike the macroscale SfM, ambient light control is the most critical factor among others in micronizing the SfM technique. Two different ambient light conditions are taken into consideration: absolute light and relative light. In the absolute light condition, the light fixture is assembled on the microscope so that a fixed ambient light condition is achieved for the photo-sequencing process. In the relative light condition, however, the light fixture is installed on the mobile platform where the target sample is in place. The concept of the absolute and relative light conditions is illustrated in Figure 5. 

Less disturbance in ambient light is anticipated in the absolute light condition for the object, while a plain diffuse light condition for the camera may be achieved by the relative light condition.

According to the paper by Triggs [24], the enabling technique for surface reconstruction by multiple 2D images is the perspective projection. Given a set of *m* projective image spaces, there is a 3D subspace of the space of combined image coordinates called the joint images, *P*^1,2,..., *n*^. These images in scaled coordinates compose a complete projective replica of the 3D world. A fundamental image in the joint image allows the reconstruction process through a matching constraint by which a set of image points, *x*^1,2,..., *m*^, are classified to be the projection of a single world point, *X*. The matching process then produces a single compact geometric object using the antisymmetric four index joint Grassmannian tensor [24]. 

The essential mechanism of projective reconstruction is matching point identification in multiple scaled images or image points, *x*^1,2,…, *m*^. A feature matching method can be used to form the fundamental matrix for the projective reconstruction. The same features in multiple images, therefore, must be easily identifiable by image transformation. 

In addition to edge or point detection, the most important aspect of the image transformation between photos are color-making attributes such as hue, intensity, lµminance, brightness, chroma, and saturation, since the same feature in different photos may not be identified due to dissimilar chromaticity. While the absolute light condition provides a stable and consistent chromaticity with respect to the microscope and, thus, to the camera, the relative light condition is anticipated to provide consistency in the chromaticity between pictures. Nevertheless, we tested two different lighting conditions to study the light effect on the microscale SfM process.

## 5. Experiments

The experimental assembly of the proposed micro-SfM system includes a gantry-type microscope for an ample space below the microscopic lens to merit the flexibility of testing different light conditions and scanning sequences. A high-definition digital camera is mounted on the microscope via a custom-made adapter for minimµm disturbance in ambient light control. 

All of the tests are performed in a light-controlled cleanroom; thus, no external light influences the photo-sequencing process other than the light fixture proposed for each configuration. The target object selected for testing is a microscale gear whose length is 300 µm, with a diameter of 70 µm (Figure 6). The surface of the gear is metal reflection; thus, the speculation reflectance may not be ignorable. Nevertheless, the object is a common industrial element good for comparison in two different ambient light control settings.

### 5.1. Absolute Light Condition

Since a 3D model by SfM is through stitching photos identified during the image-matching process, the nµmber of photos identified is the most important for 3D modeling accuracy. For fair comparison between the absolute and the relative light conditions, the same nµmber of photos (22) is taken for the projective reconstruction process. In the absolute light condition, only five photos out of 22 are identified with significant matching points in the first test (Figure 7). As a result, the reconstructed surface of the micro-gear does not represent its original cylinder shape but a crushed cylinder form (Figure 8). Through the examination of photos taken in the absolute light condition, slightly different chromaticity between neighboring photos are evidenced for the same features. Different chromaticity of the same features may cause difficulties in feature matching, thus leading to a distorted geometry of the original shape. 

### 5.2. Relative Light Condition

To the contrary, all of the photos taken in the relative light condition are identified due to the identical chromaticity of matching features in the first test (Figure 9). The localization errors of the camera viewpoints in several images in Figure 8 are due to the magnification adjustments during the photography process to obtain finer and crisper photos. The same technique is applied for the absolute light condition. As a result, the completely reconstructed micro-3D model kept the original shape intact (Figure 10). The cross-section of the gear represents a complete circle of the original gear shape with minimal distortion. Moreover, the reconstructed shape conforms to the original shape in the scale ratio; thus, virtual measurements of any part of the micro-object are made possible. For instance, we were able to measure the screw pitch to be 9.49 µm by using the reconstructed 3D model.

The success factor of the relative light condition is mainly due to the fixed natural ambient light for the sample object. This is like the case of fixing the ambient light in macroscale with the camera rotating around an object. The absolute lighting condition we tested turned out to be like the case of using a flashlight for each photograph in macroscale, thus disturbing the feature-matching process. Therefore, the discrepancy in the color attributes between potentially matching features leads to an incomplete model in SfM. Thirty times of tests for each condition revealed that the relative light condition maintained the average of 20.3 photos recognized, while the absolute light condition with 4.5 photos recognized, and with 0.2 more in the standard deviation (Figure 11).

### 5.3. Modeling Accuracy

As demonstrated in the previous section, the proposed microscale SfM revealed its capability of reconstructing the surface of a microscale object. In order to evaluate its construction capability, we tested a micro-fluidic channel to evaluate the 3D surface reconstruction accuracy in this section. The micro-fluidic channel was manufactured by using a micro-milling machine, 363-S 3-Axis Horizontal Machining Center, powered by 50,000 RPM Electric Motor Driven High-Precision Bearing Spindle. The micro-milling machine can carve a micro-shape in the scale of 50 to 100 µm with 2-micron accuracy. The width of the micro-channel created for the 3D reconstruction test was 235 µm for Lab-On-Chip applications. 

Each channel was spaced by 901 µm, and the total channel was a little less than the size of a penny (Figure 12). 

The original CAD design of the micro-fluidic channel is illustrated in Figure 12 as well. The first model was created by the magnification factor of 30 (Figure 13). The metric we used for the 3D modeling accuracy was the flatness of the top surface of the micro-fluidic channel. To that end, 17-point cloud data were captured from the model (Figure 13) and processed to create a 3D scatter plot (Figure 14). In order to measure the flatness, a plane surface fit was used to measure the deviation of each point from the fit surface. As shown in Figure 14, the Euclidean distance of each point of cloud data to the surface was measured and plotted. Most of the data points fell within the range of +0.2 to −0.2 mm. The standard deviation and the RMS value were measured to be 116 µm and 113 µm, respectively. 

In order to measure the limit of the 3D modeling accuracy of the proposed system, another model of the micro-fluidic channel was created by using the magnification factor of 70, including the digital zoom factor. As shown in Figure 15, 15-point cloud data were captured for the surface flatness accuracy test. The collected data points are illustrated in Figure 16 by scatter 3D representation and the surface fit result. As shown in the figure, all-point cloud data were within +/−0.05 mm after surface fitting, representing 35 µm of standard deviation and 29 µm of RMS value. As mentioned earlier, the proposed technique for microscale 3D reconstruction was only limited by the magnification hardware, thus less limited in object size compared to the confocal imaging-based 3D micro-reconstruction. 

### 5.4. Depth Sensing Accuracy

In order to evaluate the depth-sensing capability of the proposed micro-3D system, a micro-3D pyramid was designed and carved using the micro-milling machine. As shown in Figure 17, a 450-µm^2^ base with 2100-µm height (300 µm/step, 7 steps) micro-pyramid was created. Using the same magnitude factor (×70 zoom), a 3D model of the micro-pyramid was constructed (see Figure 18). Four-point cloud data were sampled from each level to create a plane fitting to each level. As shown in Figure 19a, a total of 12-point cloud data were depicted in a 3D scatter plot with corresponding nµmbers from Figure 18. In order to measure the height of each step, the sampled point cloud data were imported to a CAD tool to evaluate the depth measurement accuracy. For precision measurements, the fourth step of the pyramid (450 µm^2^) was used for the original and virtual pyramid calibration. The Euclidean distance of two-point cloud data samples at the outside corner of the fourth layer (1.441) was compared to the original design size of 210 µm. As shown in Figure 19b, using the calibration factor (145.73), the depth of the first two steps from the top were measured to be 287 µm and 295 µm, respectively. Therefore, the error in measurement from the original pyramid dimension was 13 µm for the first step and 5 µm for the second step, demonstrating the superior depth measurement capability of the proposed 3D micro-system. As mentioned earlier, since the 3D reconstruction accuracy of the micro-3D system was only limited by the magnification hardware, the proposed technique may achieve higher accuracy and, therefore, be less limited by the target object size.

## 6. Conclusions

A novel micro-3D object reconstruction technique was presented in the paper. Unlike the confocal imaging technique mostly for transparent objects, we proposed a SfM-based microscale 3D reconstruction technique for solid micro-objects. We studied the macroscale SfM technique and considered all of the possible aspects that needed to be reinvented in microscale. Among them were the ambient light conditioning and the photography sequencing in microscale. With the proposed photo-sequencing process, we tested two possible ambient light conditions and found that the relative light provided the better result, representing the reconstructed model close to its original shape. We found that the result was due to the fact that the relative light condition maintained the consistency in chromaticity for potentially matching features for the image stitching process.

During the series of experiments, we demonstrated 35 µm of standard deviation and 29 µm of RMS values in the surface flatness test of a reconstructed micro-surface. Assµming the measurement uncertainty was within 10% of the measurement accuracy, the proposed 3D micro-construction system demonstrated a 300-µm accuracy in the flatness measurements. In addition, the proposed system demonstrated superior depth measurement capability of the proposed 3D micro-system, with a 3D micro-pyramid specially designed and carved by a micro-milling machine. The proposed microscale 3D reconstruction technique was, first, the occlusion-free method, and second, it was free of a size limit, as long as the micropart was within the FOV (field of view) of a higher resolution microscope. In addition, compared to the confocal imaging technique, the proposed system was able to scan an object much faster than a confocal imaging system, offering an affordable solution for a broad spectrµm of solid micro-3D parts.

## Figures and Tables

**Figure 1 sensors-20-05599-f001:**
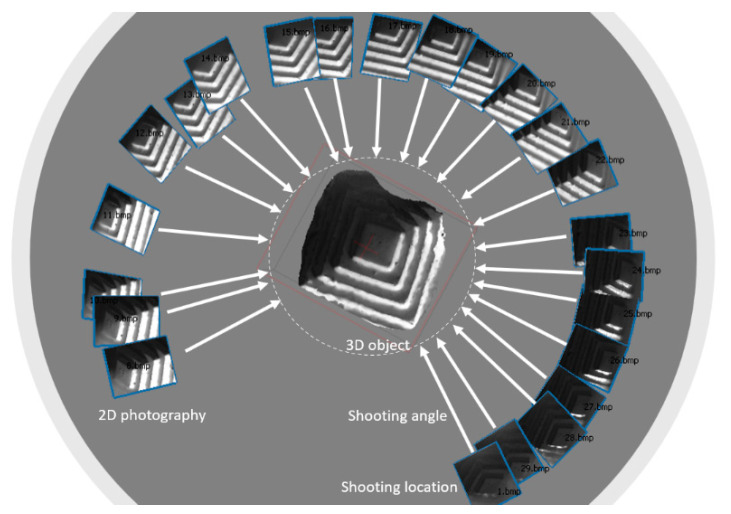
The concept of the Structure from Motion (SfM) technique.

**Figure 2 sensors-20-05599-f002:**
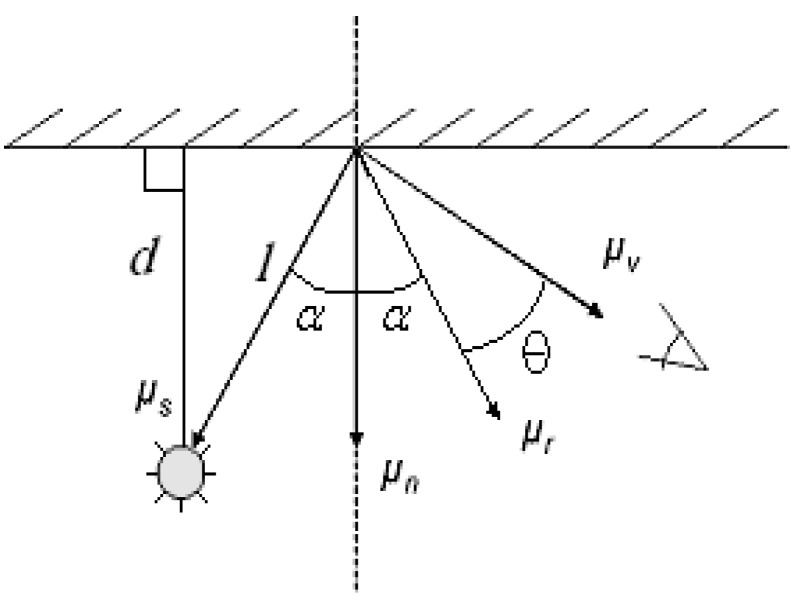
Diagram for Phong’s illµmination model.

**Figure 3 sensors-20-05599-f003:**
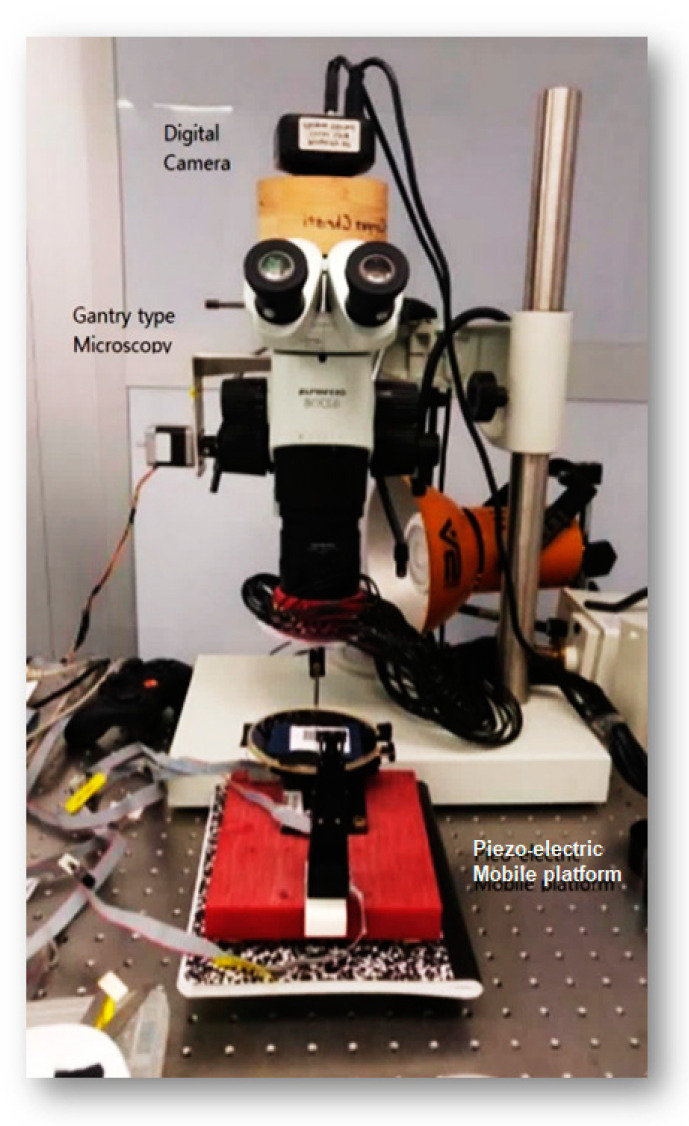
Experimental setup for microscale SfM.

**Figure 4 sensors-20-05599-f004:**
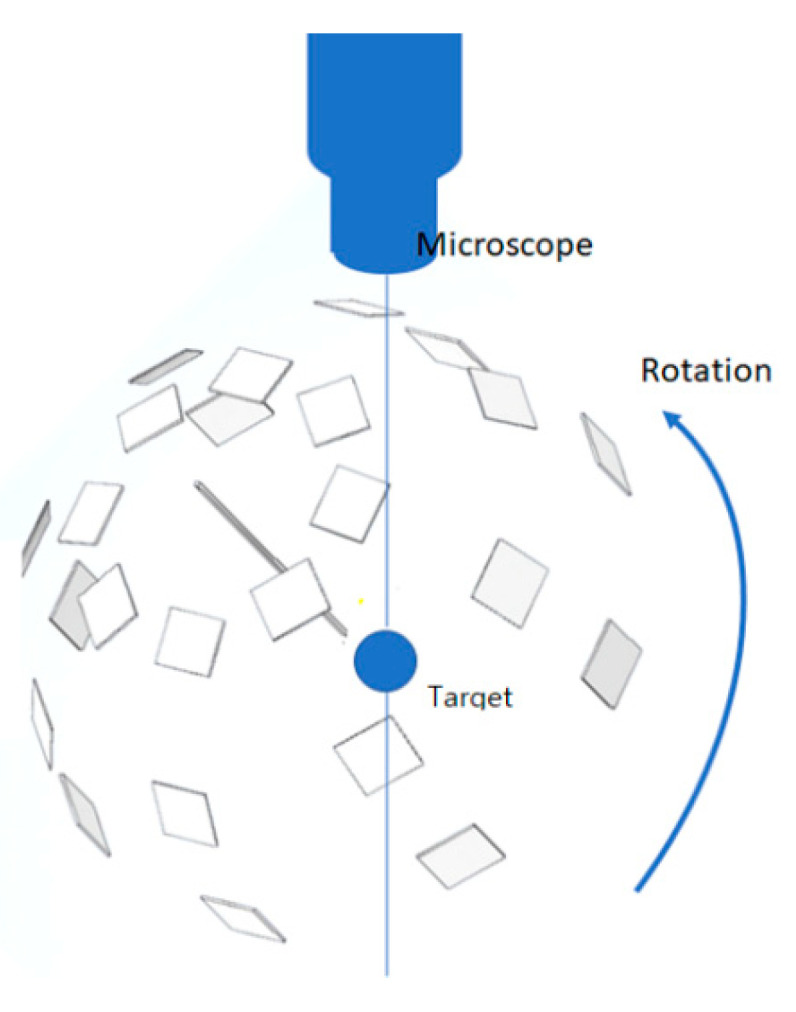
Capturing sequence for SfM in microscale.

**Figure 5 sensors-20-05599-f005:**
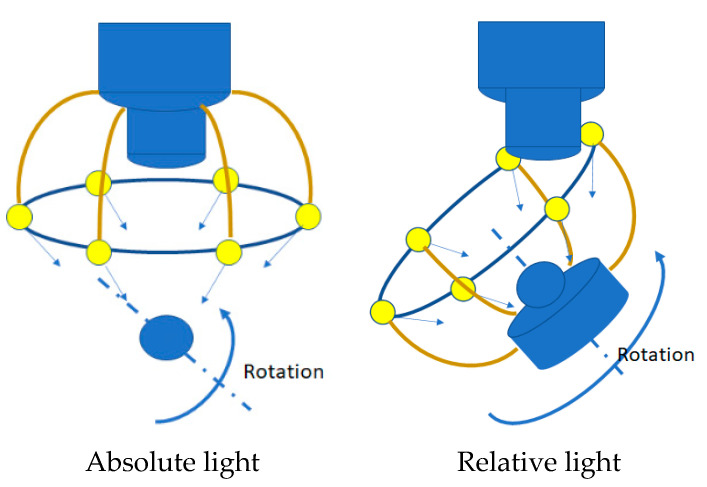
Ambient light control for SfM in microscale.

**Figure 6 sensors-20-05599-f006:**
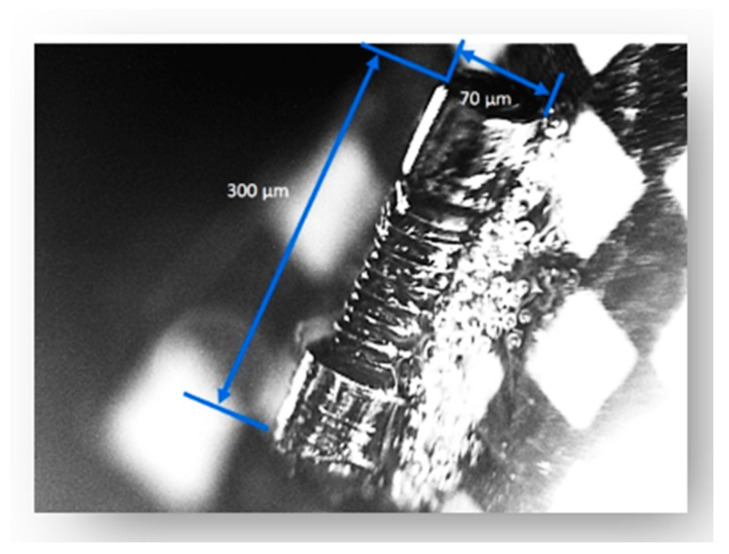
Object for experiments.

**Figure 7 sensors-20-05599-f007:**
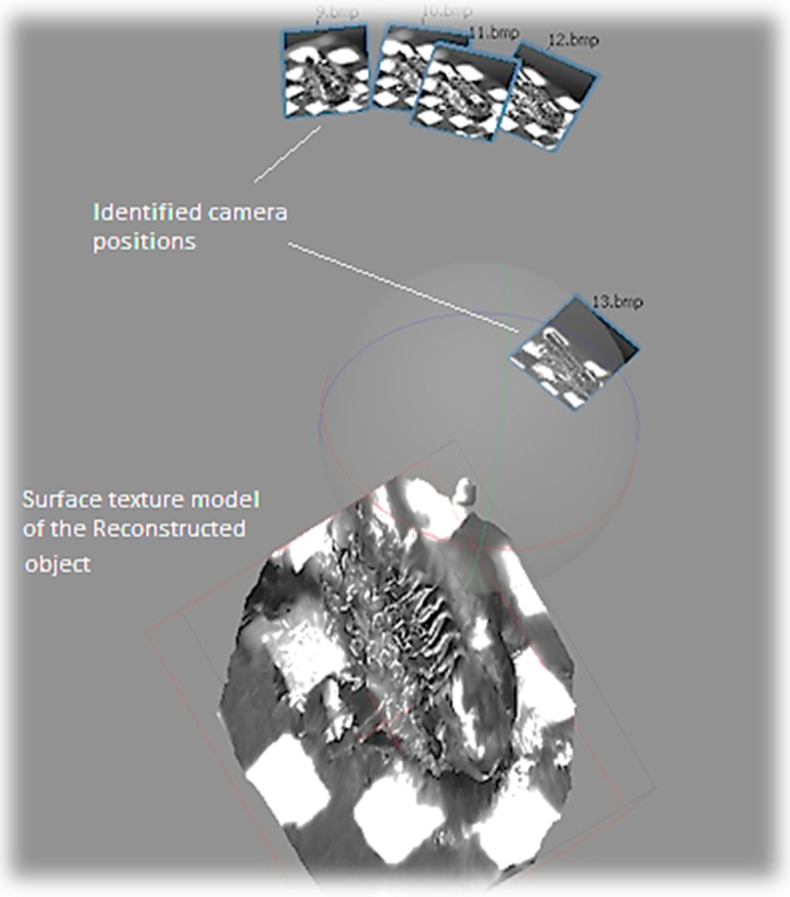
Images identified by absolute light.

**Figure 8 sensors-20-05599-f008:**
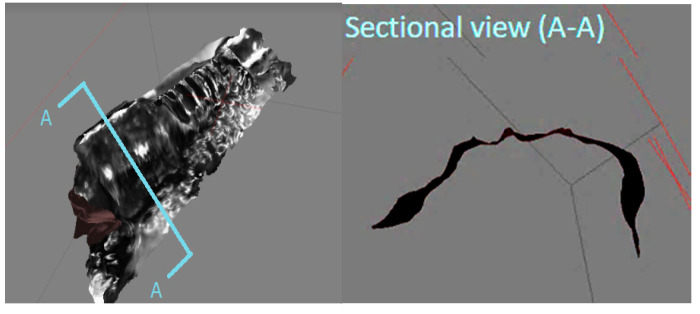
Object reconstructed by absolute light.

**Figure 9 sensors-20-05599-f009:**
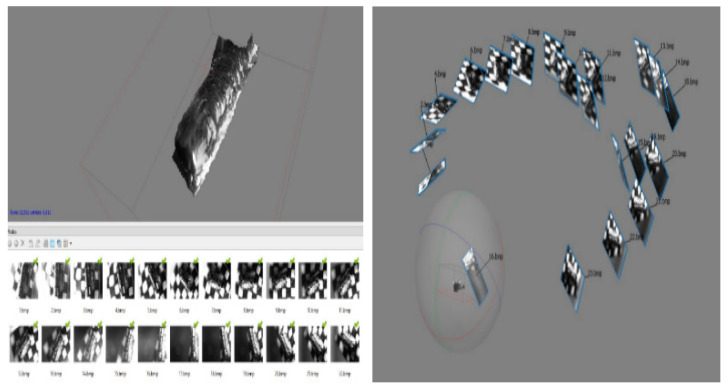
Images identified for relative light.

**Figure 10 sensors-20-05599-f010:**
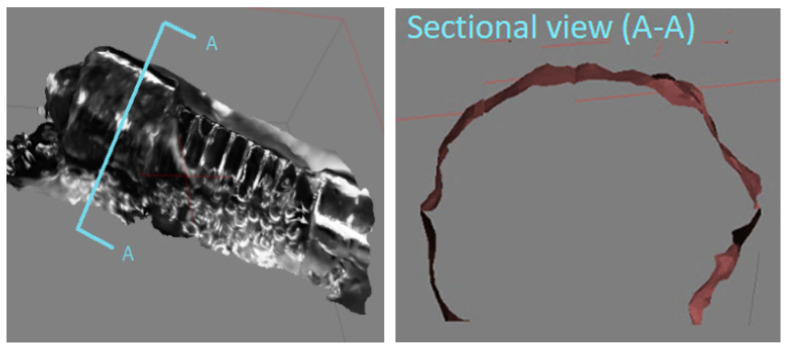
Object reconstructed by relative light.

**Figure 11 sensors-20-05599-f011:**
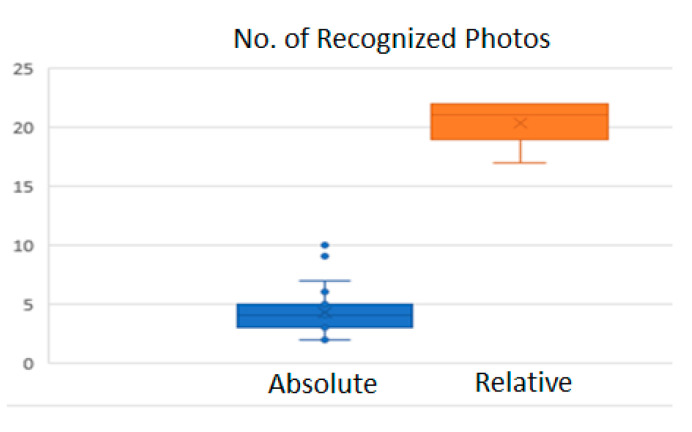
Photography recognition analysis.

**Figure 12 sensors-20-05599-f012:**
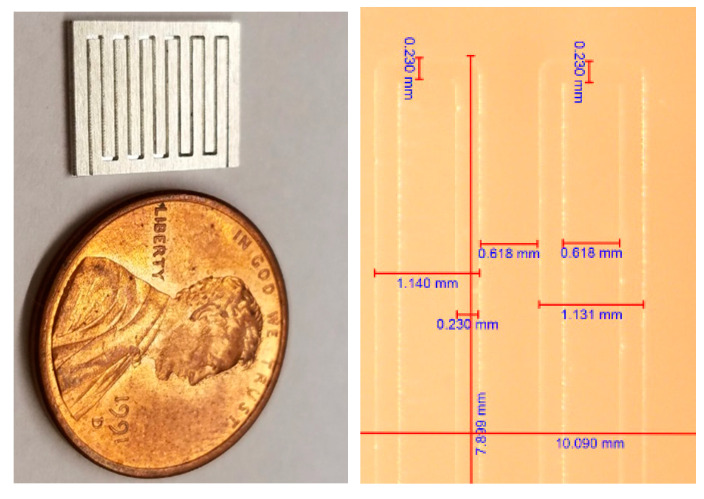
Micro-fluidic channel.

**Figure 13 sensors-20-05599-f013:**
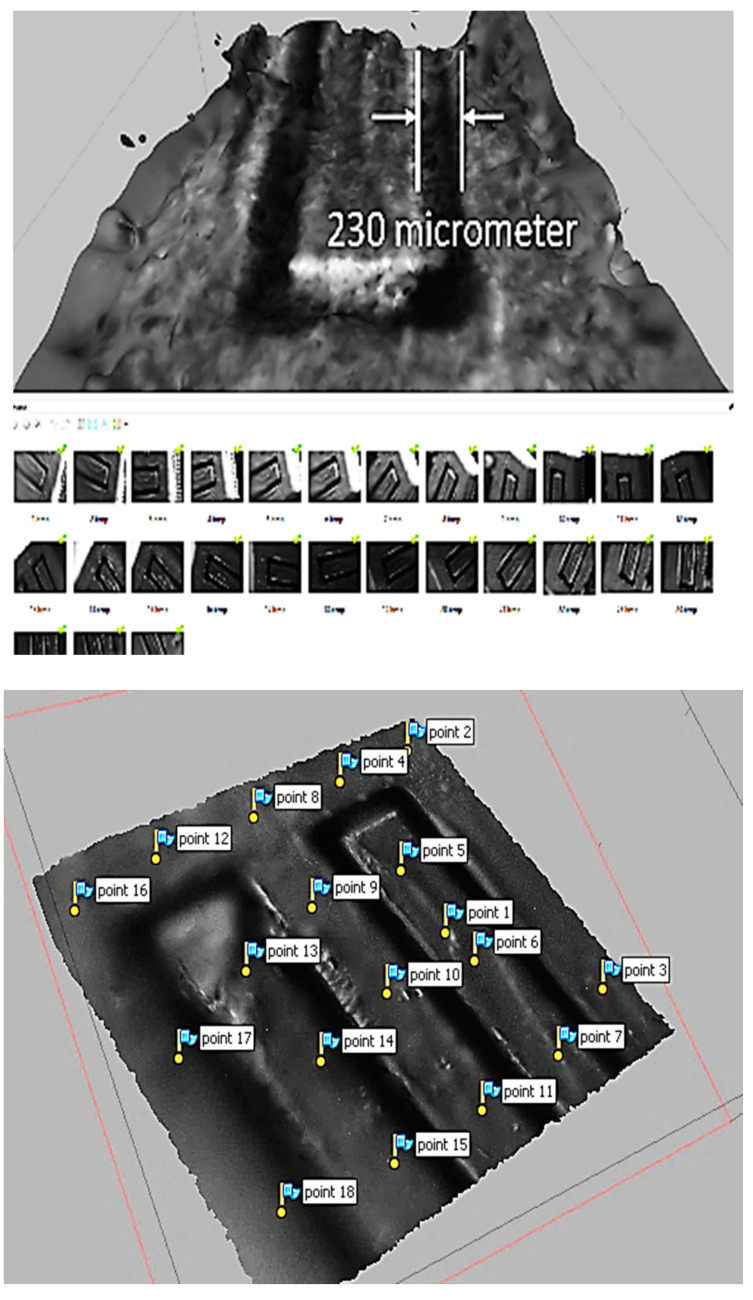
3D model of the micro-fluidic channel (×30 zoom).

**Figure 14 sensors-20-05599-f014:**
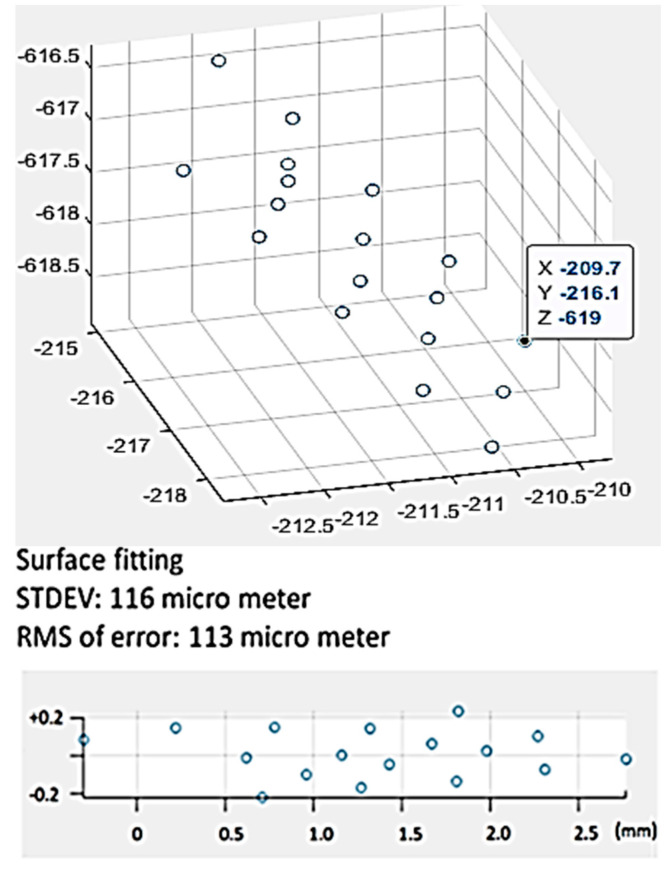
3D modeling accuracy of a surface (×30).

**Figure 15 sensors-20-05599-f015:**
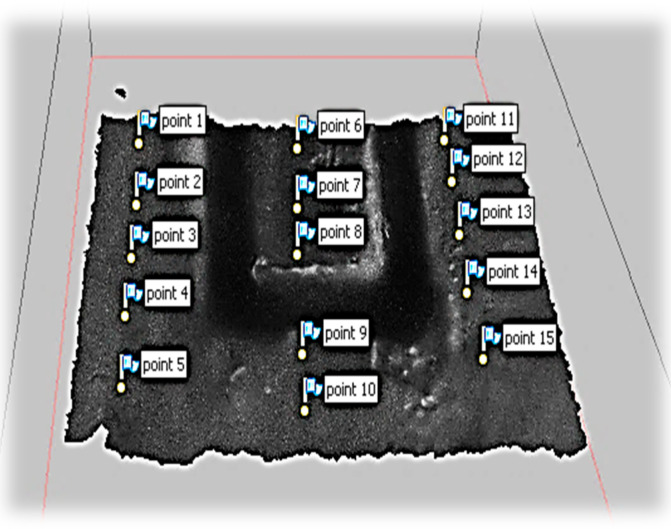
3D model of the micro-fluidic channel (×70 zoom).

**Figure 16 sensors-20-05599-f016:**
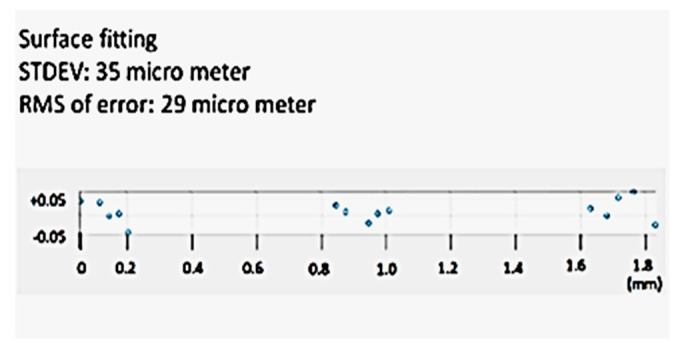
3D modeling accuracy of a surface (×70).

**Figure 17 sensors-20-05599-f017:**
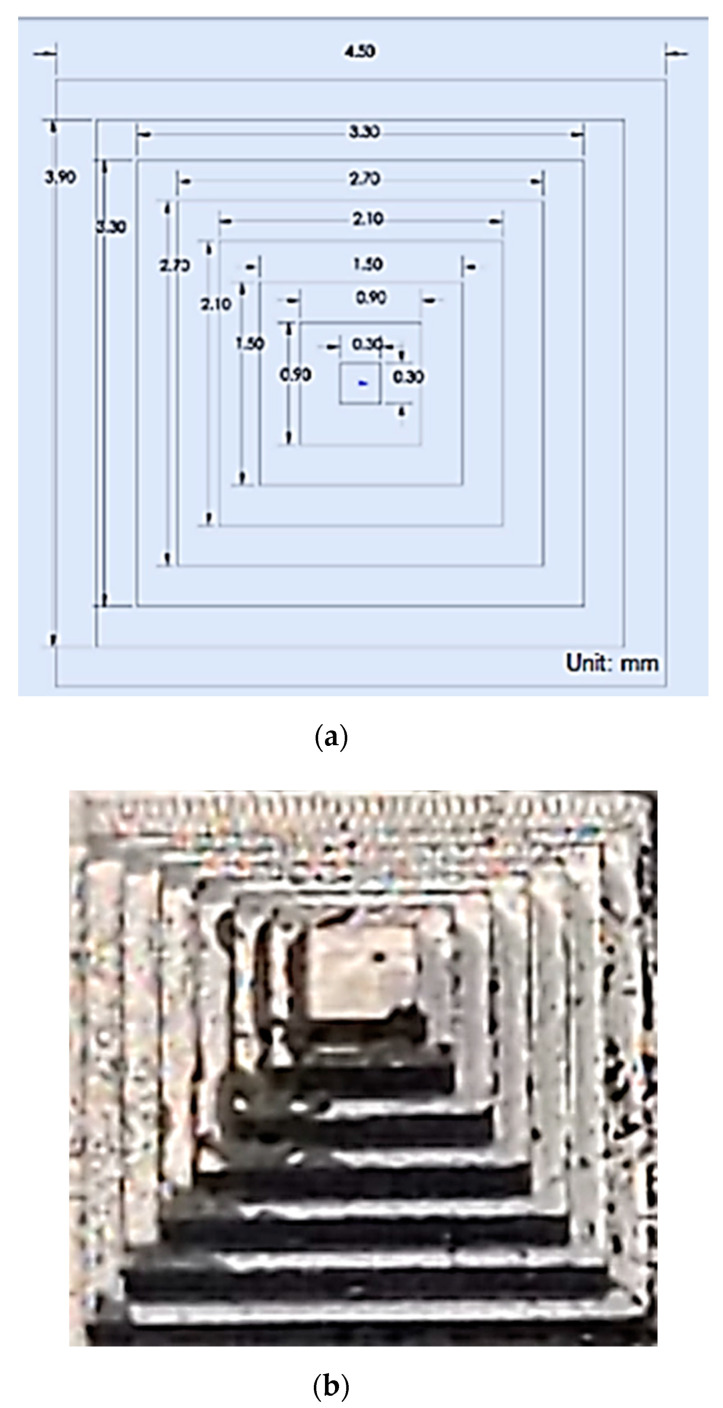
Micro-pyramid structure. (**a**) Micro-pyramid CAD design and (**b**) manufactured micro-pyramid.

**Figure 18 sensors-20-05599-f018:**
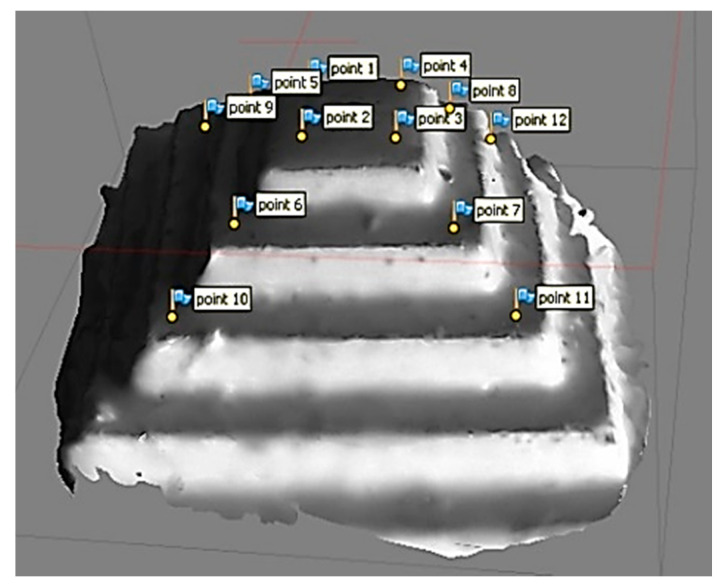
3D model of a micro-pyramid.

**Figure 19 sensors-20-05599-f019:**
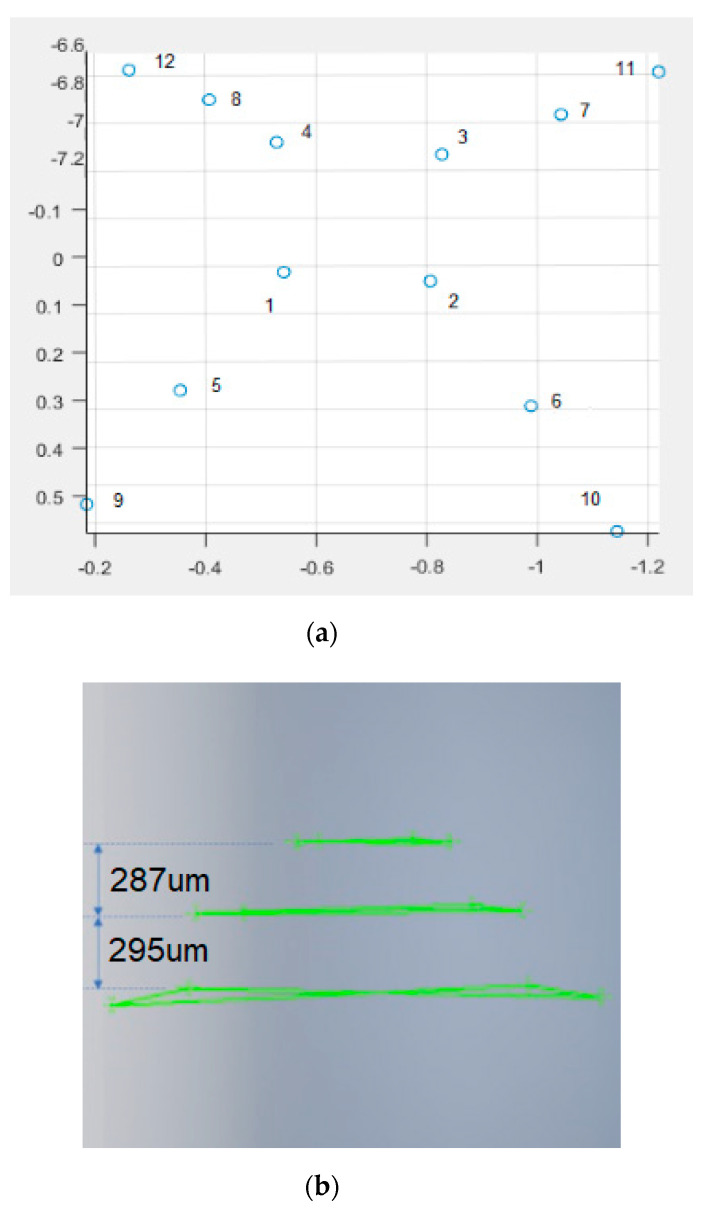
Depth accuracy measurement. (**a**) 3D scattered point cloud. (**b**) Depth measure by CAD modeling.

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
