# Peer review of "Microscopic Structure from Motion (SfM) for Microscale 3D Surface Reconstruction"

_sensors, 2020, doi:10.3390/s20195599_

Round 1
Reviewer 1 Report
In the manuscript the authors propose a novel method to construct a 3D shape in microscale for various micro sized solid objects in a broad spectrum of applications. The experiment are introduced with technical details and discussion. However, I found a very severe issue that the authors want to deal with. While, after the introduction, directly talking about the technical details, the manuscript wants to first introduce the methodology of the so called SfM.
Besides, I would like to suggest the authors paying attention to a novel imaging from motion technology. Instead of using focal plane detector array of large scale of pixels, this novel method employs only single-pixel detector, which means it has some great advantage at special wavelength. There is possibility that this SfM method could be combined and extended to the area of single-pixel imaging.
Author Response
The reviewer mentieond that there is little or no introduction for SfM technology. However, SfM is introduced in the "I. Introduction" section. Besides, I added the following paragraph to explain the SfM in more detail.
"The SfM technology reconstructs a 3D model by using motion parallax, which is the foundation of depth generation by measuring amount of move of each feature as the camera moves. For instance, an object close to camera moves faster than an object far away from it as the camera moves side by side. The fundamental mechanism of 3D construction is similar to that of the stereo-vision, but photos next to each other forms a pair of stereo-vision, thus to enable 3D reconstruction."
The second concern was about a possible R&D by merging SfM with the single pixel imaging (SPI) technology. Single pixel imaging is novel and new to authors at the moment. However, one author is very interested in the SPI and will venture a way to consider extension of the SfM with SPI in the next step of the research.
Reviewer 2 Report
The manuscript described a novel micro-surface reconstruction method using the Structure from Motion in microscale based on the photometric stereovision via microscopic photogrammetry. Generally, the manuscript is clearly structured and therefore it is easy to read and understand. The motivation and goal are specified correctly. Some minor issues need to be addressed. 1. Microscale SfM can be achieved with the improvement on scanning methodology, ambient light control, and light conditioning. Compared to a confocal imaging system, the proposed system is faster. Could the author compare the speed quantitatively? 2. What is the theoretically achievable resolution of the microscale SfM? Is it possible to use the microscale SfM for cell imaging?Author Response
In response to the reviewer's comments below, the manuscript is revised to answer the quesitons.
1. Microscale SfM can be achieved with the improvement on scanning methodology, ambient light control, and light conditioning. Compared to a confocal imaging system, the proposed system is faster. Could the author compare the speed quantitatively? 2. What is the theoretically achievable resolution of the microscale SfM? Is it possible to use the microscale SfM for cell imaging?
For the comment 1, authors added the following sentence in section III.
"In addition, compared to the confocal micro imaging technology (for instance, from Leica Microsystems), the scanning speed by the proposed Micro SfM technique is much faster. The scanning of an object with total of 30 photos by Micro SfM takes only 30sec, while the point scanning by confocal imaging with a x-y-z table for multiple images takes up to 30min or more. The fast scanning speed of the Micro SfM is because of the simple scanning procedure with 2 rotational axes of the scanning platform."
For comment 2, in section V (experiment) it is said that
"As mentioned earlier, since the 3D reconstruction accuracy of the micro 3D system is only limited by the magnification hardware, the proposed technique may achieve higher accuracy and is less limited for the target object size."
Therefore, theorectically if you use a x150 microscope, you can go down to 1-10 micrometer resolution.
Since the technique is based on stereo-vision, any object that can be created by photogramatry will be the subject of the Micro SfM. Although it is developed mainly for a micro solid object, authors feel positive that the proposed system is able to create a 3D image of a cell.
Reviewer 3 Report
In their paper, the authors present the results of a novel micro-surface reconstruction method using photometric stereovision via microscopic SfM photogrammetry. They tested two possible ambient light conditions and found that the relative light provides the better result, representing the reconstructed model close to its original shape: the result is due to the fact that the relative light condition maintains the consistency in chromaticity for potentially matching features for the image stitching process.
I have found the manuscript very interesting, but it is not organised in a clear form, and the study is not supported by enough references. In that form, the manuscript needs to be revised in order to be considered for publication.
In particular, I suggest to fix three main problems:
Firstly: The paper structure must follow the standard template: https://www.mdpi.com/journal/sensors/instructions
Reorganise your manuscript, please.
Secondly: check your reference software (Paperpile?) because something went wrong when you updated the references list. There are several “Error! Reference source not found.” along with the text. These need to be fixed.
Also:
Lines 90-91: please cite a few of these examples that you saw. Just the most significative. I don’t know if you know this paper:
Sapirstein, Philip. 2018. “A High-Precision Photogrammetric Recording System for Small Artifacts.” Journal of Cultural Heritage 31 (May): 33–45.
Anyway, there are plenty of examples on the macroscale application of SfM to digitise historical documents, architectural monuments of archaeological artefacts…
Lines 92-101: please add references to support your consideration.
Lines 109: the reference for Agisoft software is missing.
Lines 109 -112: reference missing.
Line 118: which type of microscope did you use? Please specify (in Material and Methods) type, brand and specs supported with a reference (the official manual, for example).
Lines 121 - 122: reference missing.
120-129: which type of camera did you use? Again, as for the microscope, specify (in Material and Methods) type, brand and specs supported with a reference (the official manual, for example), please.
Lines 131 -134: support your consideration with references, please.
Lines 156 - 160: that’s true, but references are missing.
Lines 160-184: references?
Lastly: the acronyms: sometimes you put the full definition after the acronyms, other times you don’t. Please be consistent along with the text. I strongly suggest to indicate the full definition for all the acronyms: you should be much clear possible because your paper could be read by non-expert audiences.
So:
- Line 31 , BRDF : indicate what it means, please.
- Line 44, MEMS: indicate what it means, please.
- Line 47, CLSM: that’s the correct way! Do it for all the acronyms, please.
- Line 51, micro-CT: I miss the significance of “CT”....
- Line 198 and Line 209, DOF: you can’t use the same acronym for two different definitions. Try to use two different forms, for example: DOFo (Depth of Focus) and DOFr (Degree of Freedom).
Author Response
Firstly: The paper structure must follow the standard template: https://www.mdpi.com/journal/sensors/instructions
Reorganise your manuscript, please.
- done
Secondly: check your reference software (Paperpile?) because something went wrong when you updated the references list. There are several “Error! Reference source not found.” along with the text. These need to be fixed
- done
Also:
Lines 90-91: please cite a few of these examples that you saw. Just the most significative. I don’t know if you know this paper:
Sapirstein, Philip. 2018. “A High-Precision Photogrammetric Recording System for Small Artifacts.” Journal of Cultural Heritage 31 (May): 33–45.
Anyway, there are plenty of examples on the macroscale application of SfM to digitise historical documents, architectural monuments of archaeological artefacts…
- Sapirstin Philip’s work along others are referenced and sited in the paper. Sentence below is added in the intro section.
“Recently several attempts are made in precision SfM for small artifacts such as a historical or cultural heritage [21][22]. In one article, 5-10 cm objects are 3D-modeled via SfM technology [21]. High resolution imagery of a 3D-printed skull (5cm) has been achieved by using a 60mm lens. Without a microscopic apparatus, however, the limit of the proposed SfM was for 5cm or larger objects due to the camera focal distance limit and interference during the photography.”
Lines 92-101: please add references to support your consideration.
- The paragraph has been revised as below.
“Although precision 3D modeling examples of a small objects (5-10cm) have been reported [21][22], a microscale SfM system has not been reported both in industry as well as in academia due primarily to problems in miniaturization. Predominantly, the SfM technique, by its nature, requires photographing an object from different angles with adequate overlaps for the stitching process. Therefore, the main hurdle toward the microscale SfM lies in the miniaturization of the SfM photographing method of a micro object.”
Lines 109: the reference for Agisoft software is missing.
- The reference is added
Lines 109 -112: reference missing.
- The reference is added
Line 118: which type of microscope did you use? Please specify (in Material and Methods) type, brand and specs supported with a reference (the official manual, for example).
- The reference added
Lines 121 - 122: reference missing.
120-129: which type of camera did you use? Again, as for the microscope, specify (in Material and Methods) type, brand and specs supported with a reference (the official manual, for example), please.
- The reference is added
Lines 131 -134: support your consideration with references, please.
- The reference is added
Lines 156 - 160: that’s true, but references are missing.
- The reference is added
Lines 160-184: references?
- The reference is added
Lastly: the acronyms: sometimes you put the full definition after the acronyms, other times you don’t. Please be consistent along with the text. I strongly suggest to indicate the full definition for all the acronyms: you should be much clear possible because your paper could be read by non-expert audiences.
So:
Line 31 , BRDF : indicate what it means, please.
Line 44, MEMS: indicate what it means, please.
Line 47, CLSM: that’s the correct way! Do it for all the acronyms, please.
Line 51, micro-CT: I miss the significance of “CT”....
Line 198 and Line 209, DOF: you can’t use the same acronym for two different definitions. Try to use two different forms, for example: DOFo (Depth of Focus) and DOFr (Degree of Freedom).
- All the acronym definitions are added and DOF (Depth of Focus) is chanced to DOFo
Round 2
Reviewer 1 Report
The authors reply to my concern saying SfM is introduced in the "I. Introduction" section. Yet I do not think this is enough for general readers to understand the technology. All through the manuscript there is no formal description/illustration of the method. The authors should understand that readers of this paper are not necessarily familiar with this technology/related area. Therefore I insist a detailed introduction besides the introduction section is necessary, and will help readers understand the work better.
Author Response
The sentence below is added as the first paragraph in section II. A reference is introduced ([29]) and a concept diagram of the SfM is created (Figure 1).
"Structure from Motion (SfM) is a 3D reconstruction technique by taking multiple 2D photography of an object from surrounding areas [29]. It is a photogrammetric range imaging technique for estimating three-dimensional structures from two-dimensional image by searching for common features or an object from different images. The assumption behind the SfM is that a near object moves more than an object far away as the camera moves (See Figure 1). The original concept of the SfM is coined by the stereo-vision studied in the fields of computer vision and visual perception. In biological vision, SfM refers to the phenomenon by which humans can perceive 3D structure from the projected 2D images at retinal areas."
Reviewer 3 Report
I am reporting just a few typos:
Line 191: "1. 1. Inability of surface texture change". Remove one of the "1."
Line 316: "3. modeling accuracy" --> capital letter missing in "Modeling".
Author Response
1. "1." is removed.
2. Capital 'M' is used.